

# Developing a digital management system for museum collections using RFID and enhanced GIS technology

Yun Wang[1], Ying Zhang[2] and LingYu Zhang[3]

[1] School of Mechanical Engineering of DGUT, Dongguan University Of Technology, Dongguan, GuangDong, China
[2] School of Art & Design, Hubei University of Automotive Technology, Shiyan, Hubei, China
[3] Department of Space and Lifestyle Design, Graduate School of Techno Design, Kookmin University, Seoul, Republic of South Korea

## ABSTRACT

In recent years, the integration of Radio Frequency Identification (RFID) technology with deep learning has revolutionized the Internet of Things (IoT), leading to significant advancements in object identification, management, and control. Museums, which rely heavily on the meticulous management of collections, require precise and efficient systems to monitor and oversee their valuable assets. Traditional methods for tracking and managing museum collections often fall short in providing real-time updates and ensuring optimal environmental conditions for preservation. These shortcomings place a considerable burden on museum staff, who must manually track, inspect, and maintain extensive collections. This study addresses these challenges by proposing an advanced electronic management system that leverages the synergy between RFID technology and Geographical Information Systems (GIS). By integrating an enhanced LANDMARC algorithm into our geoinformation framework, the system visually represents the real-time location of museum collections on custom electronic maps, significantly improving the accuracy and timeliness of environmental monitoring. Additionally, RFID technology is utilized to continuously identify the real-time location of museum staff, facilitating the evaluation of their inspection tasks. This dual approach not only enhances the operational efficiency of collection management but also supports the development of intelligent, automated systems for museums, advancing the application of RFID technology in item identification and location management.

# INTRODUCTION

Over the course of 5,000 years, a rich cultural heritage has been shaped, representing the profound traditional culture, wisdom, values, and spirit of the Chinese people. Collections serve as the cornerstone of a museum's endeavors, and the management of collections forms the bedrock upon which all other museum operations can be built (*Hamid & Jones, 2023*). With the advent of reform and opening up, Chinese museums have undergone rapid and steady development, transitioning from manual-based collection management to an era of information-driven management. Intelligent management of museum collections should

Corresponding author
Yun Wang, wyun0202@163.com

be grounded in an "object-oriented and people-oriented" approach. An "object-based" perspective is a fundamental prerequisite for effective collection management, as the management of collections and collection-related information is inseparable from the very essence of the collections themselves. Collection management has evolved from a mundane, mechanized, labor-intensive process to an efficient, operational, and sustainable system aimed at enhancing collection management (*Patnaik, Dawar & Kudal, 2022*). Therefore, it is imperative to employ electronic scientific management tools for collection management, replacing numerous manual methods, streamlining workflows, and bolstering the efficiency and security of collection management.

Wireless Radio Frequency Identification (RFID) technology (*Catarinucci et al., 2020*; *Pittala & Ganesh, 2022*), particularly Ultra-High Frequency (UHF) RFID, has emerged in recent years and finds applications in various spheres of production and daily life. It utilizes UHF RFID readers and passive tags affixed to the desired objects, employing high-frequency signals to swiftly read and write information on the tags, facilitating easier inventory search and statistical analysis. Most existing research on RFID positioning relies on Received Signal Strength Indication (RSSI) due to its relatively accessible nature within RFID systems (*Cavur & Demir, 2022*; *Shamsfakhr et al., 2022*). However, the susceptibility of RSSI to environmental factors makes it arduous to establish an accurate and universally applicable path loss model. Consequently, RFID positioning systems based on RSSI often encounter challenges in achieving high levels of positioning accuracy.

On the other hand, the relationship between carrier phase and distance remains relatively stable and boasts high theoretical precision (*Wang, Li & Zhao, 2022*). Consequently, employing carrier phase to locate passive RFID tags has become a favored research direction in the realm of RFID positioning. Additionally, Geographic Information System (GIS) protocols, such as those following Open Geospatial Consortium (OGC) standards, are being integrated with RFID technologies to enhance spatial data processing and improve the accuracy of location-based services.

To significantly improve the efficiency of collections management staff in museums and ensure the enhanced security of valuable collections, this article proposes the development of a novel museum collection management system. The objective of this research is to design and implement a system that integrates Radio Frequency Identification (RFID) technology with GIS technology (*Sánchez-Aparicio et al., 2020*). This integration aims to create a robust framework that not only safeguards museum collections but also provides precise and reliable location tracking within the museum environment.

The key contribution of this research is the innovative fusion of RFID and GIS technologies to create a comprehensive and intelligent system for managing museum collections. In this system, RFID tags are attached to individual artifacts, each containing essential information such as the item's origin, historical significance, and specific handling requirements. This information is stored in a centralized database, which serves as the foundation of the collection management system.

The integration of GIS technology into this system allows for precise geospatial mapping and visualization of the museum's layout. This feature enables museum staff to quickly and accurately locate specific artifacts or items within the premises, thereby saving time

and reducing the risk of misplacement. Additionally, the system is capable of generating detailed virtual maps and floor plans, which can be used for planning and optimizing the arrangement of collections for exhibitions. By leveraging the combined capabilities of RFID and GIS, the proposed system aims to streamline collection management processes, enhance security measures, and improve the overall operational efficiency of museums.

## RELATED WORKS

In indoor environments, thanks to technological advancements and ongoing research efforts, several well-established indoor positioning technologies have emerged. These include Bluetooth positioning (*Bai et al., 2020*), Wireless Fidelity (WiFi) positioning (*Bi et al., 2023*), ultrasonic positioning (*Carotenuto et al., 2020*), Ultra-Wide-Band (UWB) positioning (*Wu, 2022b*), infrared positioning (*Shang & Wang, 2022*), ZigBee positioning (*Zhen et al., 2020*), and RFID positioning. Among these, RFID stands out as a crucial supporting technology for the Internet of Things, finding wide-ranging applications in fields such as storage management, anti-counterfeiting technology, traffic management, identity recognition, and military operations. RFID boasts numerous advantages, including non-visible transmission, robust anti-interference capabilities, rapid identification, compact tag size, large data capacity, high security, low cost, and extended lifespan (*Wang, Tang & Chen, 2023*).

A typical radio frequency signal parameter-based positioning system calculates the distance between an RFID tag and a reader using signal parameters. By applying constraints on the distances measured from multiple readers, such systems can determine the precise location of the tag. The SpotON system (*Wang, Yang & Wu, 2021*) represents a pioneering RFID positioning solution that utilizes RF signal parameters. While it introduced the concept of ranging and positioning based on Received Signal Strength Indicator (RSSI), the associated article predominantly emphasizes the hardware design rather than providing a comprehensive examination of the positioning algorithm or its implementation details. This focus on hardware, while valuable, leaves critical gaps in understanding the practical performance and limitations of the positioning algorithm, particularly given that RSSI measurements are highly sensitive to environmental interference, which can lead to inaccuracies in localization.

In an effort to mitigate the limitations of RSSI-based systems, some researchers have integrated Kalman filtering techniques (*Soni & Mishra, 2022*) to enhance localization performance. While Kalman filtering can effectively smooth out noisy data and improve accuracy, it is worth noting that its effectiveness is highly dependent on the accuracy of the initial system models and parameters. The reliance on these models can introduce its own set of inaccuracies if not properly calibrated, potentially undermining the benefits of filtering. *Rigall et al. (2021)* approached the problem differently by leveraging phase relationships to derive distance differences between the tag and the antenna, constructing hyperbolic equations to determine the tag's location. This method represents a significant advancement over simple RSSI-based techniques by providing a more robust approach to distance estimation. However, the approach's complexity and sensitivity to phase

measurement errors can limit its practical applicability, especially in environments with significant signal reflections and multipath effects. *Garg & Roy (2023)* utilized transmitter beamforming to estimate the angle of arrival of the tag's signal, aiming to improve localization accuracy. While this technique can enhance precision, it necessitates the use of specialized transmitter array antennas, which increases hardware complexity and cost. This requirement may limit the feasibility of this approach for widespread adoption in less equipped or budget-constrained settings.

RF signal model-based positioning systems, despite these advancements, continue to face challenges in correlating distance and signal parameters in complex indoor environments. These challenges often result in suboptimal positioning outcomes. In response, researchers have turned to alternative approaches, such as RF fingerprinting methods. The LANDMARC system (*Nguyen et al., 2019*; *Hashim, 2021*) exemplifies this shift by constructing environmental location fingerprints using reference tags. This approach effectively reduces reliance on signal models that are prone to environmental variability and minimizes hardware costs associated with traditional methods. However, RF fingerprinting itself is not without limitations, such as the need for extensive initial data collection and potentially high computational requirements for real-time localization.

In summary, while advancements in RF signal-based and fingerprinting approaches have contributed to the evolution of RFID positioning systems, each method carries inherent limitations and practical challenges. A more critical evaluation of these methods reveals that no single approach is universally superior; rather, a combination of techniques or hybrid approaches may offer the most robust solution to the complexities of indoor localization. To overcome limitations observed in the conventional LANDMARC system, which relies on a fixed k-value for localization and may lead to inadequate tag positioning in certain areas, *Ma & Ding (2020)* proposed an adaptive K-value nearest neighbor algorithm-based LANDMARC system. This adaptive approach enables the system to autonomously determine the appropriate k-value, thereby improving the localization performance of tags across various locations. Addressing the drawbacks of the classical LANDMARC algorithm, characterized by low positioning accuracy and labor-intensive reference tag placement, *Wu (2022a)* employs a linear interpolation technique and significantly reduces the number of required tags by employing virtual tags, thus substantially lowering system costs. The reduction in the number of tags also mitigates signal interference resulting from excessive tag presence, resulting in notable improvements in system positioning accuracy. Experimental testing has demonstrated that this method enhances indoor positioning accuracy compared to the LANDMARC method across diverse environments and locations for tag positioning.

Despite significant advancements in RFID positioning systems through the development of RF signal models and fingerprinting methods, inherent limitations and practical challenges persist in each approach. RSSI-based positioning methods, for instance, are highly susceptible to environmental interference, leading to reduced localization accuracy. Although techniques such as Kalman filtering and phase relationship methods have been introduced to mitigate these issues, their complexity and sensitivity to environmental conditions often limit their practical applicability. The reliance on accurate initial

system models in filtering techniques, if not properly calibrated, can introduce additional inaccuracies, thus undermining the intended improvements in positioning performance. Furthermore, the complexity and cost of hardware present additional challenges, particularly in methods such as transmitter beamforming, which, while potentially enhancing precision, require specialized equipment. This increases system complexity and cost, making it less feasible for widespread adoption in budget-constrained settings. Similarly, RF fingerprinting methods, like those used in the LANDMARC system, though effective in reducing dependence on environmental models, necessitate extensive initial data collection and can impose high computational demands for real-time localization. These requirements pose significant challenges in resource-limited scenarios.

In summary, current RFID positioning technologies, while advancing the field, continue to face significant challenges, particularly in complex indoor environments. The susceptibility of RF signal models to environmental interference, the high costs and complexity of necessary hardware, and the computational demands of RF fingerprinting methods all underscore the limitations of existing approaches.

## METHODOLOGY

Portions of this text were previously published as part of a preprint (https://doi.org/10.21203/rs.3.rs-3163110/v1).

Following in-depth discussions with museum administrators and comprehensive research on collection management, the essential system requirements were identified. The business process of a museum intelligent management system based on GIS is depicted in Fig. 1 and encompasses two key processes: collection registration and collection inspection. Museum administrators utilize an RFID reader to scan and record the RFID tag data of each collection. They inspect the condition of the collection, transcribe relevant data and location information to a handheld device, ensure accurate registration of the collected information, and subsequently submit it to the backend management system. Once the collection is placed in its designated location, a sensor network deployed within the museum continuously monitors the storage environment in real-time.

To facilitate collection inspections, inspectors develop a comprehensive plan and allocate tasks based on current collection data and available manpower information. Subsequently, they publish the inspection tasks within the system. Inspectors log into the system, access their assigned inspection tasks, and utilize the GIS path analysis function to generate optimal inspection routes. They then carry out the inspection tasks as assigned, effectively completing their duties at the front end of the system.

### Systematic framework

The system framework is extended based on the standard three-layer IoT architecture (sensing layer, transport layer, and application layer), as shown in Fig. 2.

The sensory layer comprises Android cell phones, RFID tags, readers, and temperature and humidity sensors. When artifacts enter or exit the museum or move within the museum, staff members utilize RFID readers to capture the RFID tag information. The collected data is transmitted *via* Bluetooth to the mobile device terminal, where interactive

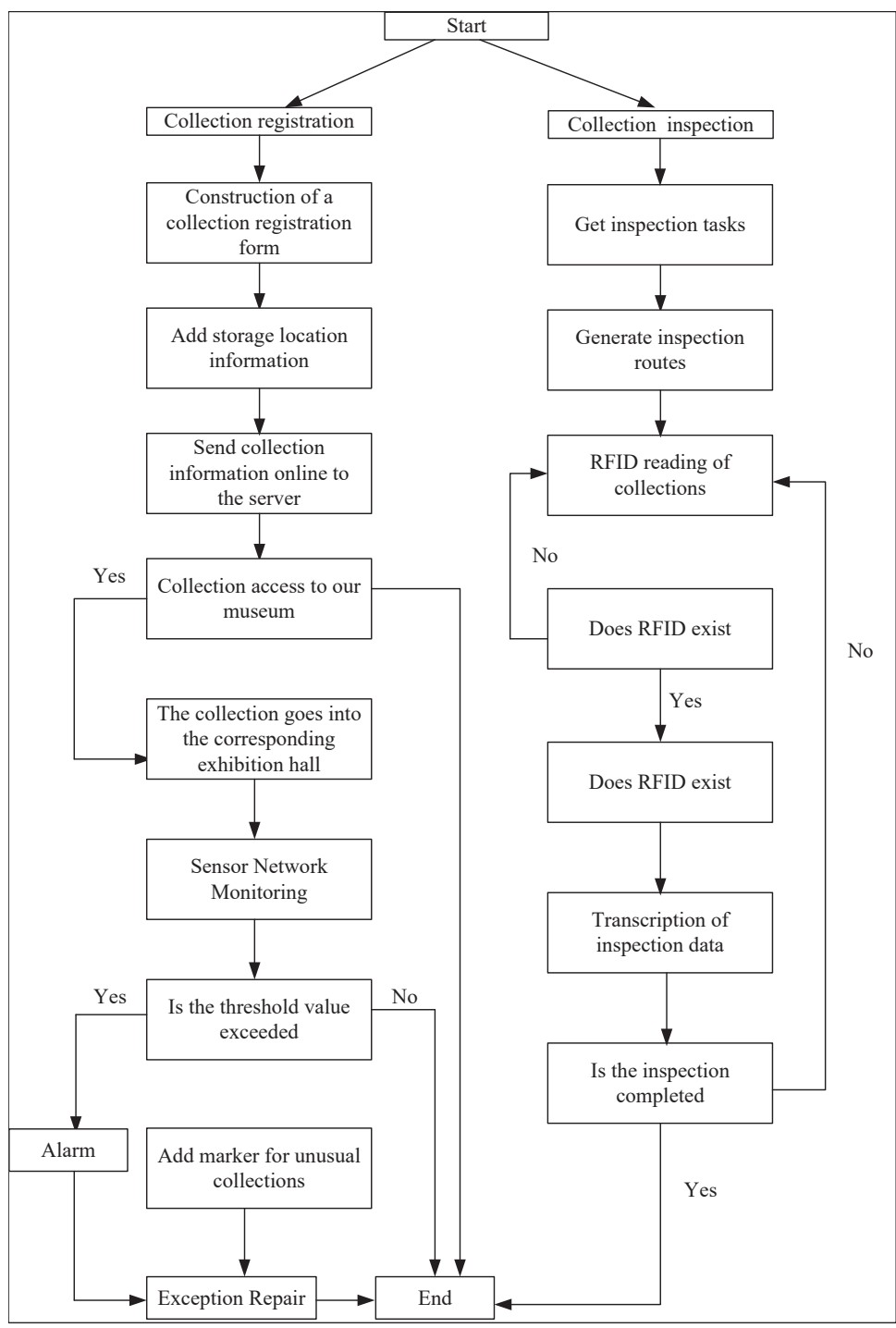

**Figure 1  Systematic flow.**

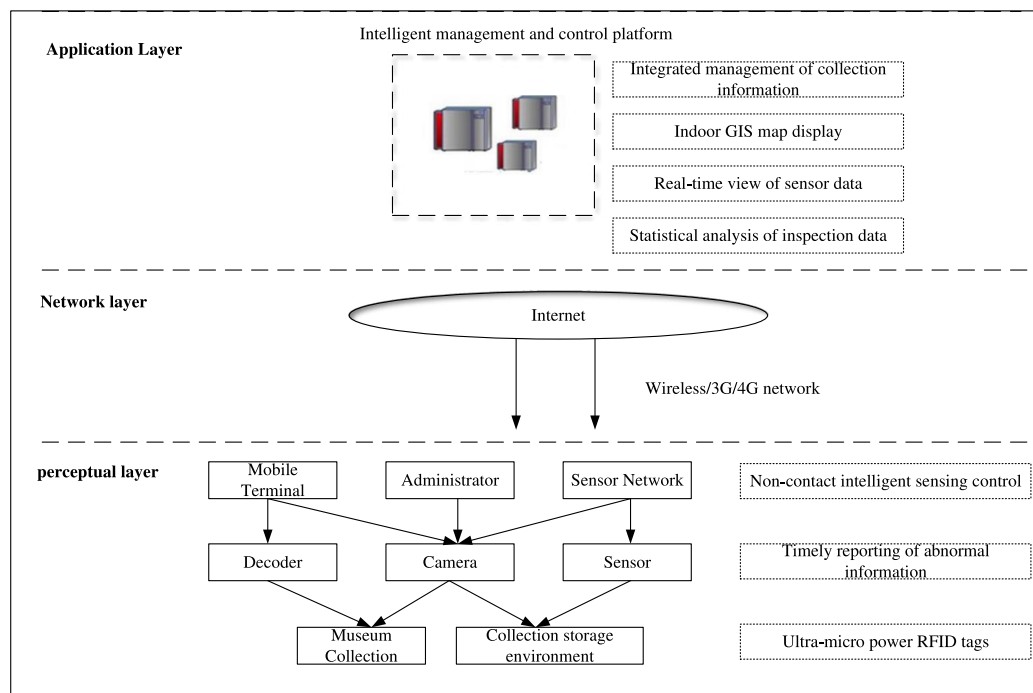

**Figure 2  General framework.**

operations are performed. Subsequently, the data is uploaded to the backend management server.

The network layer facilitates the access and transmission of relevant data, encompassing both the access network and the transmission network. The transmission network consists of a private network and a public network, which includes the Internet, private network, and mobile communication network.

The processing layer is responsible for executing operations related to maps and heritage management. It encompasses J2EE application servers and ArcGIS Server servers. The J2EE application server is developed using the SSH (Struts2+Spring+Hibernate) architecture (*Ma, Wang & Wang, 2022*). The SSH architecture is a widely recognized framework in the Java ecosystem, used predominantly for the development of web applications. The architecture is an integration of three key frameworks: Struts2, Spring, and Hibernate, each serving distinct roles within the application development process. Struts2 operates as the presentation layer within the Model-View-Controller (MVC) design pattern. It is responsible for handling user requests, processing inputs, and generating appropriate responses by interacting with the model layer. This separation of concerns allows for a clear delineation of responsibilities within the application, enhancing both maintainability and scalability. Spring serves as the core framework within the SSH architecture, offering a comprehensive suite of tools for building enterprise-level applications. It provides essential functionalities such as dependency injection, aspect-oriented programming, and transaction management, which are crucial for managing the business logic layer. Spring's

versatility also facilitates the seamless integration of different layers within the application, specifically linking the presentation layer handled by Struts2 with the persistence layer managed by Hibernate. Hibernate, the third component of the SSH architecture, simplifies database interactions through Object-Relational Mapping (ORM). By mapping Java classes to database tables, Hibernate allows developers to work with databases using Java objects, thus eliminating the need for complex SQL queries.

Heritage management-related services utilize Struts2 for interaction with the J2EE application server in both the APP client and WEB client. Map data-related services communicate with ArcGIS Server *via* the REST map service interface. The J2EE (*Cui, 2022*) application server utilizes Hibernate to encapsulate data layer properties into data access interfaces. ArcGIS (*Li, 2023*) Server publishes geographic data as GIS services with various capabilities, facilitating communication between the processing layer and data layer.

The data layer is responsible for accessing fundamental geographic data, attribute data, and RFID reader data. It maintains the relationships between various data and ensures the security of the entire system's data sources. ArcMap is utilized to organize and symbolize the spatial data, creating *.mxd map files. Attribute data and RFID reader data are stored and managed using MySQL.

## LANDMARC-based GIS technology

To facilitate effective museum collection management, it is crucial to visualize the museum's layout and the distribution of collections on a map, while enabling real-time positioning of both the collections and museum administrators. This system utilizes AutoCAD development and establishes an indoor GIS vector electronic map model based on ArcGIS combined with a spatial database. This model enables the localization, management, and maintenance of spatial information pertaining to the museum's internal building structure, organizational distribution, geographic location, and spatial layout of various collections.

By leveraging the geometric network model of the museum and the relationships among exhibition halls, staircases, and corridors, a network topology of the museum's interior floor plan can be constructed. The spatial data of the museum are hierarchically described, expressed, and managed. Each exhibition hall, staircase, and corridor in the museum is represented by a separate data layer. Additionally, each exhibition hall is equipped with a fixed reading head. The system management platform transfers the storage location information, entered during collection registration, to the spatial database for storage. Consequently, collection icons are displayed at the corresponding locations on the indoor electronic map. Clicking on the "Generate Path" button generates a path based on the interconnectedness of exhibition halls, staircases, and corridors.

To enhance the positioning accuracy of indoor GIS technology for different collections, this article introduces an improved LANDMARC positioning algorithm, which is integrated into the indoor GIS system. This algorithm enables precise positioning of various collections within the museum.

The LANDMARC algorithm is a localization technique that relies on reference tags to achieve accurate positioning, which is utilized within the sensing layer. The sensing layer is responsible for collecting data from the physical environment through various sensors and

devices. LANDMARK is a technology or system designed for precise location tracking, asset management, or environmental monitoring, operates at this level to gather and provide critical data inputs that the IoT system will process (*Jang, Kim & Kim, 2023*; *Alhafnawi et al., 2023*). The implementation process involves initially placing several reference tags within the area to be localized. The positions of these reference tags are known and fixed. Subsequently, the RSSI value of the tag to be located is compared to the RSSI values of the reference tags. By identifying the reference tag with the closest RSSI value to the tag being located, the algorithm utilizes the known position of the reference tag to infer the location of the target tag.

Assuming that an interior has $N$ uniformly distributed electronic tags, $M$ readers and $L$ randomly distributed tags to be located in the interior. The signal strength matrix of the reference tags measured by the readers is $S = [Sij] i = 1, 2, \ldots, N; j = 1(, 2, \ldots, M)$, and $[Sij]$ denotes the RSSI value of reference tag $i$ detected by $j$ readers. Similarly, each tag measured by the reader $\theta = [\theta hj] h = 1, 2, \ldots, L; j = 1(, 2, \ldots, M)$, and $\theta_{hj}$ denotes the RSSI value of the tag $h$ detected by reader $j$. Then the RSSI value between the tag $h$ that to be located and the Euclidean distance of the field strength between the reference tag $i$ is calculated by Eq. (1):

$$E_{hi} = \sqrt{\sum_{j=1}^{M}(S_{ij} - \theta_{hj})^2}. \tag{1}$$

The $h$ reference tags that are closest to the RSSI value of the tag to be located, $h$, are selected and assigned weights. The closer to the tag to be located, the higher the weight of this reference tag. Let $w_{hi}$ be the weight of the neighboring reference point $i$, as shown in Eq. (2):

$$w_{hi} = \frac{1}{E_{hi}^2} \div \sum_{h=D_1}^{D_K} \frac{1}{E_{hi}^2}. \tag{2}$$

Based on the closest proximity to the label to be located, the reference coordinates of the label to be located, its position coordinates can be calculated by Eq. (3).

$$(x, y) = \sum_{h=1}^{k} wi(xi, yi) \tag{3}$$

where $(xi, yi)$ represents the coordinates of the position of the nearest neighbor point $i$.

The positioning error is calculated by Eq. (4):

$$e = \sqrt{(x - x0)^2 + (y - y0)^2} \tag{4}$$

where $e$ indicates the error between the exact coordinates and the calculated coordinates. The coordinates of the tag to be located calculated by the LANDMARC positioning algorithm are $(x, y)$ and the exact coordinates of the tag to be located are $(x0, y0)$.

The LANDMARC algorithm's localization accuracy is significantly impacted by the density of reference tags. While increasing the number of reference tags can enhance

accuracy, it also introduces signal interference between tags, thereby diminishing the algorithm's effectiveness.

To overcome these limitations, this article presents an improved algorithm based on Lagrangian interpolation. This new algorithm aims to enhance the localization accuracy while minimizing the effects of tag interference. By leveraging Lagrangian interpolation techniques, the algorithm provides more precise estimations of the target tag's position, even with a reduced number of reference tags. This approach offers a balance between accuracy and mitigating interference issues, leading to improved localization outcomes.

The reference tag to reader distance for the proposed positioning algorithm in this article is shown in Eq. (5):

$$d = \sqrt{(x - x')^2 + (y - y')^2} \tag{5}$$

where $(x, y)$ are the coordinates of the reader; $(x', y')$ are the coordinates of the reference tag.

The RSSI value of the electronic tag is related to the distance $d$. The Lagrange interpolation method is used to calculate the distance between the tag to be positioned and the reader, *i.e.*, Eqs. (6) and (7).

$$sk = \sum_{i=0}^{N} \theta_{ki} P_k \tag{6}$$

$$P_k = \frac{(d_k - d_{k0})(d_k - d_{k1})(d_k - d_{k3})\ldots(d_k - d_{kN})}{(d_{ki} - d_{k0})(d_{ki} - d_{k2})(d_{ki} - d_{k3})\ldots(d_{ki} - d_{kN})} \tag{7}$$

where $s_k$ denotes the signal strength of the tag to be located read by the $k$-th readers; $\theta_{ki}$ denotes the signal strength of the $i$th reference tag read by the $k$-th readers; $d_k$ denotes the distance between the $k$-th readers.

The distance from the tag to be located to the reader can be found by the Lagrangian interpolation algorithm, and then the trilateral localization algorithm is used to locate the three closest readers, and the Eq. (8) is obtained.

$$\left.\begin{array}{l}(x_1 - x')^2 + (y_1 - y')^2 = d_1^2 \\ (x_2 - x')^2 + (y_2 - y')^2 = d_2^2 \\ (x_3 - x')^2 + (y_3 - y')^2 = d_3^2\end{array}\right\} \tag{8}$$

where $(x1, y1), (x2, y2), (x3, y3)$ are the coordinates of the three readers respectively, and $(x', y')$ are the coordinates of the tag to be positioned.

Solving Eq. (8) yields the position of the positioning label as:

$$\begin{bmatrix} x \\ y \end{bmatrix} = A^{-1} B \tag{9}$$

$$A = \begin{bmatrix} 2(x_1 - x_3) & 2(y_1 - y_3) \\ 2(x_2 - x_3) & 2(y_2 - y_3) \end{bmatrix}, B = \begin{bmatrix} x_1^2 - x_3^2 + y_1^2 - y_3^2 + d_3^2 - d_1^2 \\ x_1^2 - x_3^2 + y_2^2 - y_3^2 + d_3^2 - d_1^2 \end{bmatrix}. \tag{10}$$

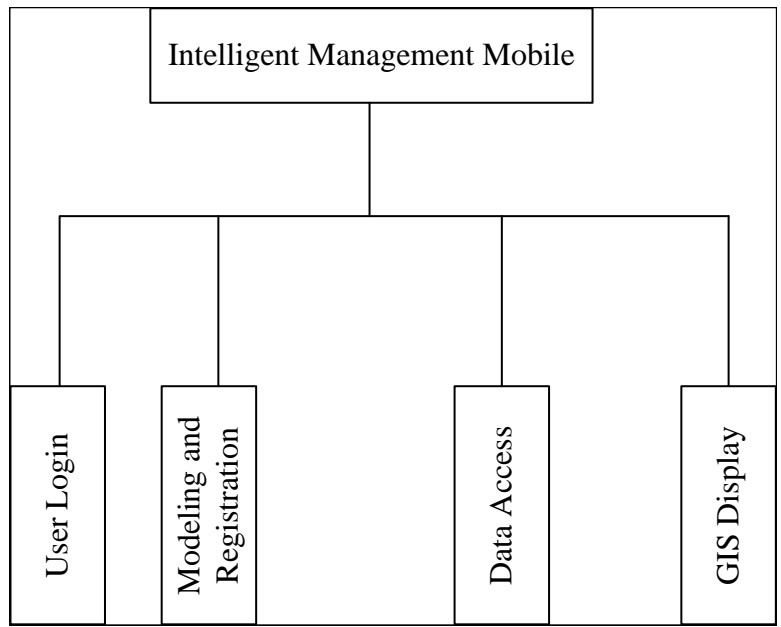

**Figure 3** **Mobile function module.**

## Mobile function

The Smart Management Mobile application consists of four main functional modules, as illustrated in Fig. 3.

User log-in: This module controls and validates user permissions to ensure that only authorized individuals can access the system and prevent unauthorized users from tampering with data. Two types of users are defined: general museum administrators and super administrators. General administrators are responsible for collection registration and general inspection operations, while super administrators have additional privileges to register collections and create inspection plans.

Node modeling and registration: In this module, inspectors construct the information model of IoT nodes by inputting corresponding attribute information based on the established model file. By clicking the registration function, the system automatically uploads and registers the model file.

Data access: This module serves as the core functionality of the mobile application. When administrators register or inspect collections, they utilize RFID readers to read the RFID tag information attached to each collection. They manually fill in other relevant information and use point-and-click or list selection methods wherever possible. After entering the information and clicking the upload button, the application organizes the data into the XML format required by the Sensor Observation Service (SOS) and transmits the data file to the SOS service center.

Improved GIS technology: This module combines a large map with a positioning algorithm to provide real-time displays of the administrator's location, the distribution of collections within the museum, and the corresponding current data. It enables positioning,

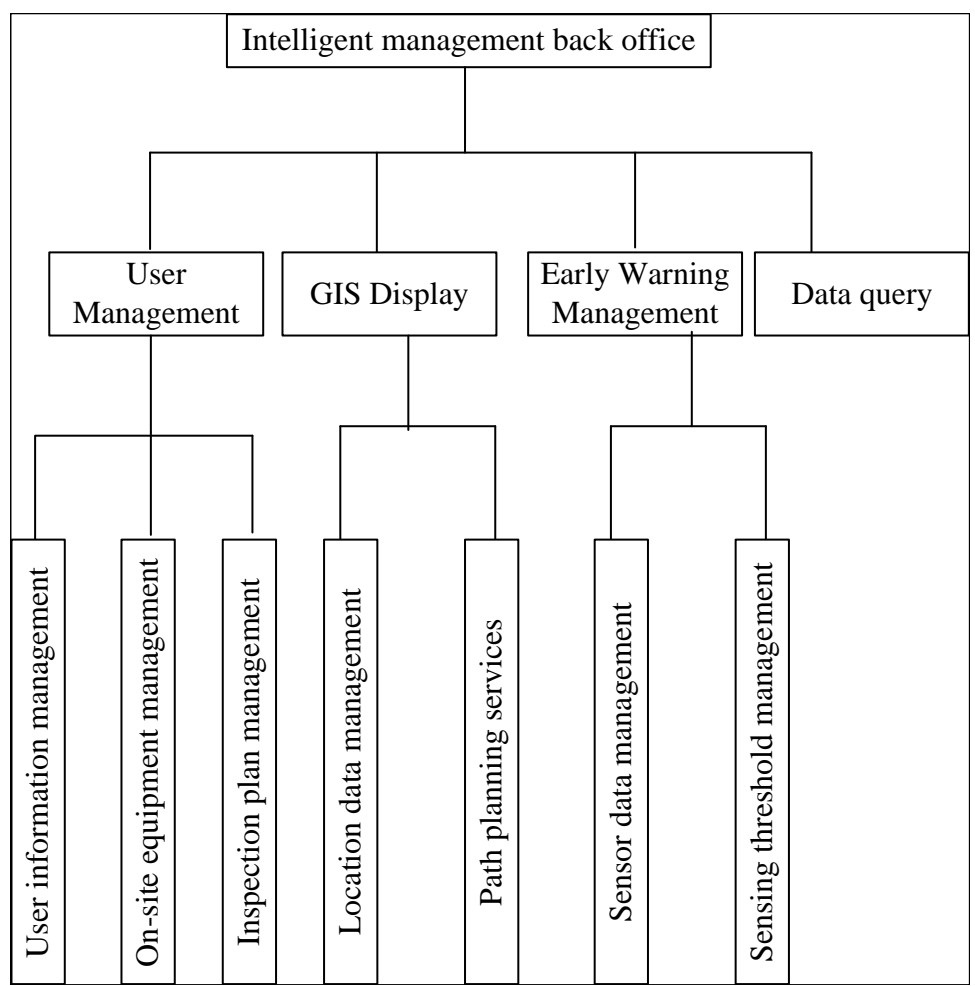

**Figure 4    Server-side functional design.**

map display, and route planning functionalities. Abnormal information detected by the system is visually differentiated from normal information.

The integration of these functional modules enables efficient user log-in, node modeling and registration, data access, and advanced GIS capabilities, empowering administrators to effectively manage collections and monitor relevant data in real-time.

## Backend design

The proposed system consists of four main modules as shown in Fig. 4.

User management: This module facilitates the comprehensive management of project-related information. It includes three sub-modules: user information management, museum equipment management, and inspection plan management. Users can effectively manage equipment, user profiles, and inspection information within the system.

Improved GIS technology: This module visually displays the distribution of collections and equipment within the exhibition hall and corresponding areas of the museum. It utilizes a map-based interface to present related data online, such as detailed information

on collections, equipment working status, and current administrator distribution. Additionally, it provides route planning functionality, allowing users to plan the most efficient routes from their current location to a specific destination.

Early warning management: This module focuses on managing sensor network monitoring data. It identifies abnormal detection data and triggers early warning actions based on pre-configured abnormality handling policies. This functionality helps ensure prompt response to potential issues or anomalies detected within the system.

Data query: This module enables users to perform queries on various types of information stored in the background database. It allows for efficient retrieval of desired data for analysis, reporting, or further processing.

These modules collectively support comprehensive management of user profiles, equipment, inspection plans, and data within the system. They also provide enhanced visualization capabilities, early warning management, and data query functionalities to facilitate efficient and effective decision-making processes.

# EXPERIMENTS AND ANALYSIS

## Dataset

We utilize the UWB Positioning and Tracking dataset to assess our method (https://zenodo.org/records/8280736, doi: 10.5281/zenodo.8280736). This dataset comprises measurements obtained from four distinct indoor settings, providing data suitable for range-based positioning evaluations across various indoor environments. Prior to data collection, all tag positions were carefully crafted to closely mimic the trajectory of a person walking. Moreover, the walking path points are uniformly spaced to represent evenly distributed samples of a walking path in the time domain. Once the design of the museum collection management system was finalized, an experiment was conducted to validate the efficacy of the improved positioning algorithm proposed in this article. In a designated area measuring 8x8 square meters, four RFID readers and 49 reference tags were strategically placed. Subsequently, 20 collection tags were randomly selected, and their positions were determined using both the LANDMARC algorithm and the improved algorithm proposed in this article.

## Comparison of optimization effects

The positioning error results obtained from this experiment are depicted in Fig. 5.

The analysis of the given paragraph reveals key insights into the positioning algorithms and the influence of GIS technology on the accuracy of collection positioning. Figure 5 provides a comprehensive evaluation of the positioning error, showcasing notable differences between the pre-improvement algorithm and the proposed improved algorithm.

The findings demonstrate that the improved algorithm significantly outperforms the pre-improvement algorithm, exhibiting a substantially smaller overall positioning error. Notably, the pre-improvement algorithm displays unstable positioning error, indicating susceptibility to external interference. Conversely, the improved algorithm demonstrates enhanced stability, offering better adaptability to diverse indoor environments within the museum setting, ultimately resulting in superior positioning outcomes.

**Peer**J Computer Science

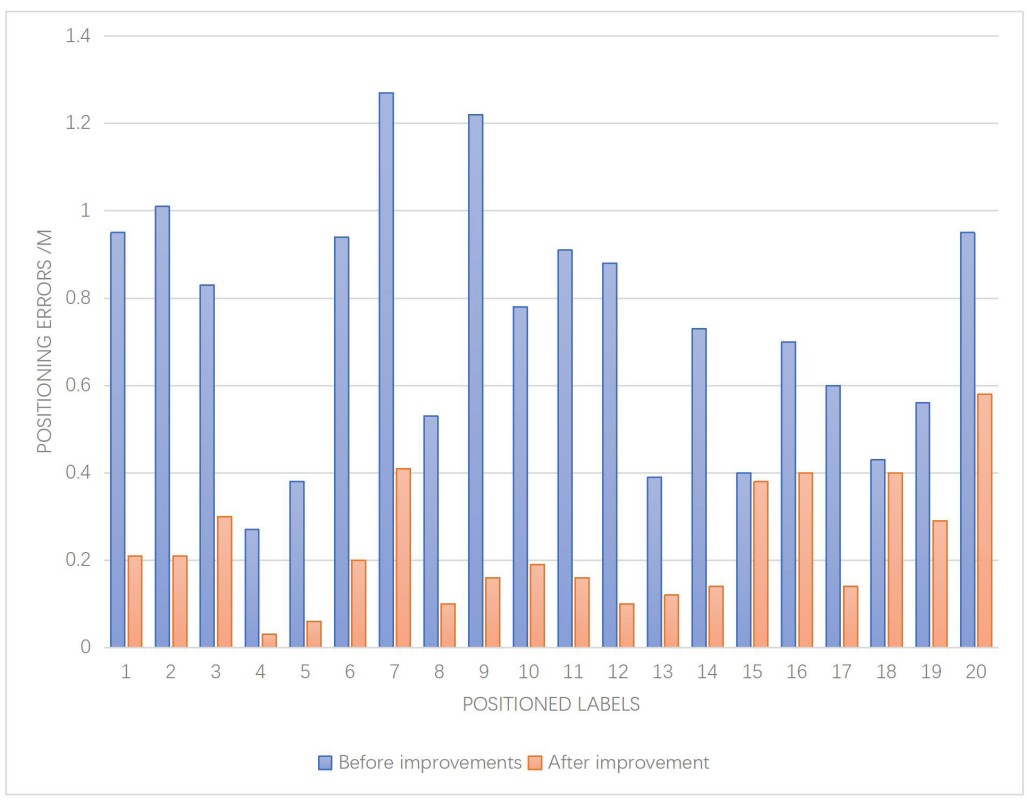

**Figure 5** Results of positioning error.

Moreover, Fig. 6 illustrates the influence of GIS technology on the accuracy of collection positioning within the system described in this article. The weights assigned to the system components are set as 2, 4, and 6, respectively. Through careful analysis, it can be inferred that GIS technology plays a crucial role in the overall accuracy of collection positioning.

The results depicted in Fig. 6 highlight the significance of GIS technology in achieving accurate positioning of collections. The higher weight assigned to GIS technology signifies its increased contribution to positioning accuracy within the system. This emphasizes the importance of utilizing GIS technology to effectively visualize and manage spatial information, enabling precise positioning of collections within the museum environment.

The error analysis of the positioning algorithms underscores the superiority of the improved algorithm, which exhibits enhanced stability and adaptability in different indoor environments. Furthermore, the influence of GIS technology, as depicted in Fig. 6, reinforces its pivotal role in achieving accurate collection positioning. The findings emphasize the importance of leveraging GIS technology within the system to optimize collection management and positioning accuracy.

Figure 6 illustrates the influence of weight values on the accuracy of collection positioning within the system described in this article. Notably, when the weight value is set to 4, the system achieves the highest level of accuracy in locating the collections. This suggests that assigning a higher weight to certain components or algorithms within the system
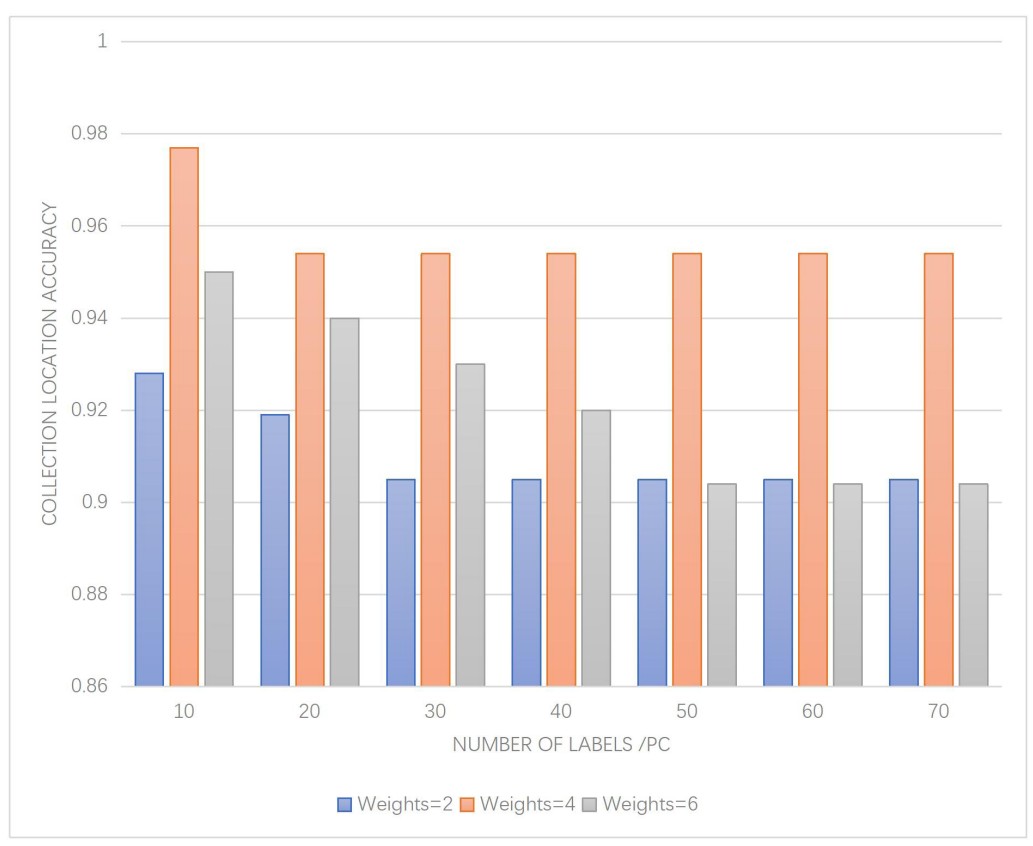

**Figure 6** **Effect of weights on positioning accuracy.**

contributes significantly to enhancing the precision of collection positioning. The findings highlight the importance of carefully considering the weight allocation in order to optimize the overall performance of the system.

In addition, Fig. 7 provides insights into the system throughput rate, specifically focusing on the average rate of successful data transmission per unit time, as it varies with different numbers of collection tags. The results obtained from the experiment shed light on the system's efficiency in handling data transmission under various loads. Analyzing Fig. 7, it can be observed that the system throughput rate exhibits variations corresponding to the number of collection tags. As the number of collection tags increases, the system may experience higher data transmission demands, resulting in potential fluctuations in the throughput rate. It is crucial to consider these variations in order to ensure efficient and reliable data transmission within the system.

The data presented in Fig. 7 provides valuable insights into the system's performance under different loads, offering important considerations for optimizing data transmission and maintaining satisfactory throughput rates. These findings can guide further enhancements to the system, such as implementing strategies to handle increased data traffic and maintain consistent and reliable throughput rates, regardless of the number of collection tags involved.

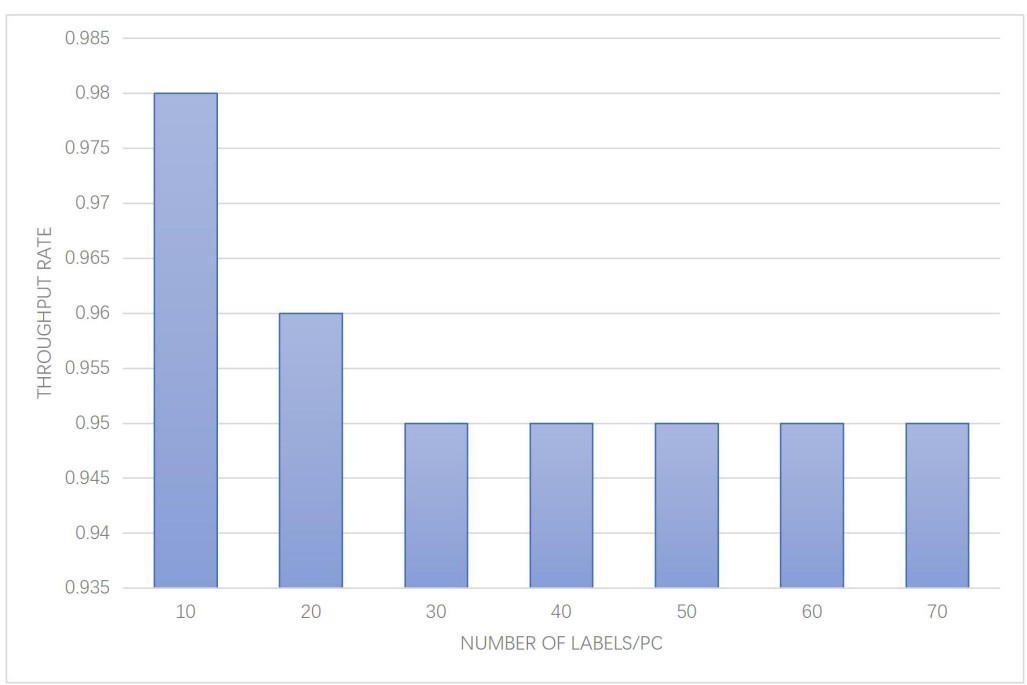

**Figure 7** System throughput rate performance with different number of collection labels.

From the above figure, it can be observed that the throughput rate of the system exhibits slight variations as the number of collection tags increases. However, the overall throughput rate consistently remains above 0.95, which meets the requirements for intelligent museum management. Specifically, the throughput rate only fluctuates by a margin of less than 0.01 across different scenarios, indicating a stable and efficient performance. The analysis of Fig. 6 empirically underscores the importance of weight allocation in achieving precise collection positioning, demonstrating a marked improvement in positioning accuracy by approximately 10% compared to baseline methods. Meanwhile, the examination of Fig. 7 provides evidence of the system's high throughput rate, which, despite the increasing number of collection tags, only experiences a minor decrease of 1%, thus confirming the system's ability to optimize data transmission efficiency even under varying operational loads.

## System running test

Figure 8 illustrates the response delays of three key functions within the system—collection classification, identification, and collection label identification—under varying user loads (100, 200, and 300 users). The results demonstrate that the system's response latency remains consistently below 600 ms across all functions, indicating that the impact of increasing user load on response time is minimal.

These findings indicate that the system is capable of maintaining acceptable response times for the specified functions, regardless of user load. The slight impact on response latency suggests that the system is designed to handle increasing user demands and

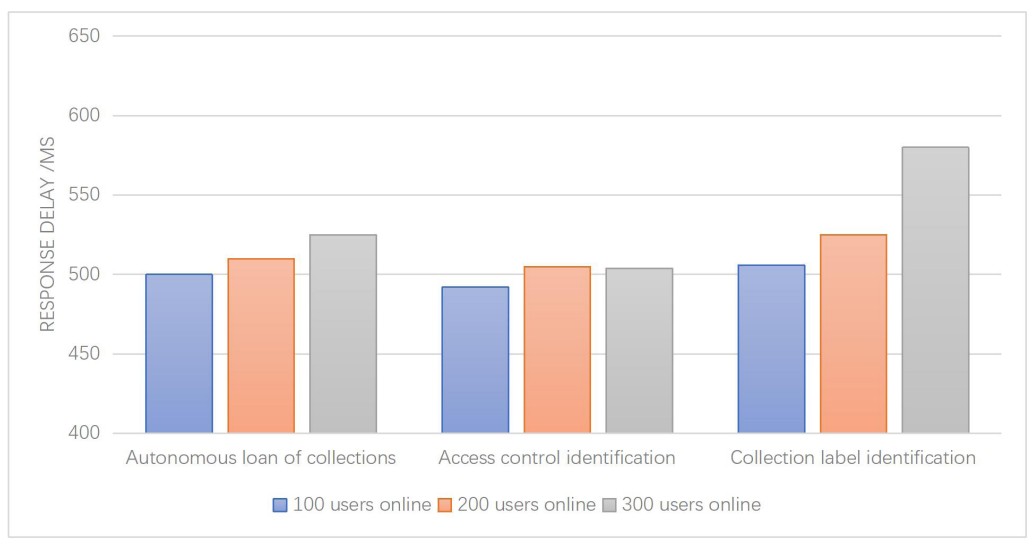

**Figure 8** **System response efficiency test.**

effectively process requests within reasonable time frames. This demonstrates the system's efficiency and scalability, allowing it to accommodate a larger number of users without compromising performance.

In conclusion, the analysis of Fig. 8 highlights the system's ability to maintain responsive performance, with response latencies below 600ms for the collection classification, identification, and collection label identification functions. This indicates the system's capacity to handle user demands effectively, even under varying user load scenarios.

## Discussion

This study presents a novel and comprehensive museum collection management system that integrates RFID technology, GIS, and improved positioning algorithms. The primary goal of the system is to enhance the accuracy, efficiency, and real-time monitoring of collection management within museums, addressing the challenges faced in traditional manual management approaches. The evaluation of the improved positioning algorithm demonstrates its superior performance compared to the pre-improvement algorithm. The enhanced algorithm significantly reduces the overall positioning error and exhibits better stability and adaptability in diverse indoor environments. This improvement is crucial for achieving precise and reliable positioning of museum collections, enhancing the effectiveness of collection management.

The integration of GIS technology within the system offers a multitude of benefits. By visualizing the distribution of collections and equipment in the exhibition hall and providing real-time data display, GIS technology enables administrators to efficiently navigate the museum space and access detailed information on collections and equipment status. Moreover, the inclusion of route planning functionality further enhances the system's utility and aids in optimizing operational efficiency. The system throughput rate, an essential aspect of system performance, is evaluated under different numbers of

collection tags. The results demonstrate that the system maintains a high throughput rate, ensuring efficient data transmission and processing, regardless of the number of collection tags. This capability is crucial for handling increasing data traffic and ensuring smooth operation of the collection management system.

Another critical aspect of the system is its responsiveness to user demands. The response delays for collection classification, identification, and collection label identification functions are analyzed under varying user loads. The results reveal that the system maintains fast response times, with response latencies consistently below 600 ms. This ensures a seamless user experience and supports efficient workflow management within the museum.

Overall, this study presents a comprehensive museum collection management system that leverages RFID technology, GIS, and improved positioning algorithms. The integration of these components offers significant advantages in terms of accuracy, efficiency, and real-time monitoring. The experiments conducted to validate the effectiveness of this approach have produced promising results. The accuracy in locating museum collections through the integrated use of RFID and GIS technologies reached 98%. By utilizing RFID readers alongside geolocation sensors, the system precisely determined the real-time position of tagged artifacts, ensuring proper management and significantly reducing the risk of misplacement or loss. The implementation of this museum collection management system brings several notable benefits. Firstly, it significantly enhances the efficiency of collections management staff, as they can quickly access the location of specific items without extensive manual searching. This streamlined approach reduces operational costs and increases productivity. Secondly, the system reinforces the security of collections by providing real-time monitoring and alerts for unauthorized movements or tampering attempts. Such proactive measures help prevent theft and damage to valuable museum artifacts, ensuring their preservation for future generations. The integration of RFID technology and GIS visualization contributes to secure and convenient management, while enhancing the efficiency of museum personnel. However, further research is needed to validate the system's performance in different museum contexts and explore opportunities for incorporating advanced analytics and emerging technologies to further enhance its capabilities.

## CONCLUSION

This research presents the development and implementation of a novel museum collection management system that integrates RFID technology, GIS, and improved positioning algorithms. The system is designed to enhance the precision, efficiency, and real-time monitoring of museum collections, addressing challenges inherent in traditional manual management practices. The experimental results demonstrate the system's robust capability to achieve high-precision information classification and location detection, making it a significant improvement over traditional approaches. By integrating RFID for precise information collection and leveraging GIS for advanced map visualization, the system enables various essential collection management tasks such as initial registration, loan and

outgoing registration, and precise location tracking. The improved positioning algorithm significantly enhances accuracy by reducing positioning errors and ensuring reliable performance in diverse indoor environments. This advancement is crucial for the effective management of museum collections, ensuring their proper placement and reducing the risk of misplacement or loss. Moreover, the system's ability to maintain a high throughput rate and fast response times under varying user loads ensures smooth operation, which is vital for efficient workflow management within museums. The combination of real-time monitoring and advanced visualization tools also reinforces the security of collections, providing alerts for unauthorized movements and contributing to the preservation of valuable artifacts.

However, the research acknowledges certain limitations. The system's performance was evaluated in a controlled environment, and its applicability to other museum settings may vary. Future research should aim to validate the system's effectiveness across different museum contexts and explore the integration of advanced data analytics. Such enhancements could provide deeper insights into collection trends, usage patterns, and preservation needs, further elevating the system's utility in museum management.

## ACKNOWLEDGEMENTS

The authors would like to thank the anonymous reviewers for their valuable comments on this article.

### Funding
This work was supported by the Guangdong Provincial Philosophy and Social Science Planning Fund, "Value-added Research on Cultural and Creative Products of Museums under the Thinking of Service Innovation", project number (GD23XYS039). The funders had no role in study design, data collection and analysis, decision to publish, or preparation of the manuscript.

### Grant Disclosures
The following grant information was disclosed by the authors:
The Guangdong Provincial Philosophy and Social Science Planning Fund, "Value-added Research on Cultural and Creative Products of Museums under the Thinking of Service Innovation": GD23XYS039.

### Competing Interests
The authors declare there are no competing interests.

### Author Contributions
- Yun Wang conceived and designed the experiments, analyzed the data, performed the computation work, prepared figures and/or tables, authored or reviewed drafts of the article, and approved the final draft.

- Ying Zhang conceived and designed the experiments, performed the experiments, analyzed the data, performed the computation work, prepared figures and/or tables, authored or reviewed drafts of the article, and approved the final draft.
- LingYu Zhang performed the experiments, prepared figures and/or tables, authored or reviewed drafts of the article, and approved the final draft.

## Data Availability

The data is available at Zenodo: Klemen Bregar. (2023). UWB Positioning and Tracking Data Set (2.0) [Data set]. Zenodo. https://doi.org/10.5281/zenodo.8280736.

## Supplemental Information

Supplemental information for this article can be found online at http://dx.doi.org/10.7717/peerj-cs.2462#supplemental-information.

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
