# Peer review of "Developing a digital management system for museum collections using RFID and enhanced GIS technology"

_PeerJ Computer Science, doi:10.7717/peerj-cs.2462_

## Round 0.1 · original submission · Minor Revisions

Dear authors,

Thank you for your submission, our experts has evaluated your work and they feel that they work is okay but needs couple of major changes to be incorporated before we reconsider it. I do agree with them and also suggests the following changes to be incorporated before you submit a detailed revised paper.

Editor Comments

The specific objectives of the study could be more clearly stated
a more detailed comparative analysis. Including specific benchmarks, metrics, and real-world data to demonstrate the system’s superiority over traditional methods could strengthen the validity of the claims

It would be valuable to discuss the practical challenges of implementing such a system in museums, such as the cost of installation, technical training for staff, and system maintenance etc

The language must be polished professionally

Reviewer 1 ·

Basic reporting

The paper presents a novel approach by integrating RFID technology with geographical information systems (GIS), offering a comprehensive and advanced solution for museum collections management. This fusion of technologies enhances the system's precision and real-time capabilities, which is a significant advancement over traditional methods. However, the incorporation of the following suggestions will bring more quality to the paper.
1. Please re-write the title, perhaps starting with an action word will be more appropriate
2. The abstract must introduce the problem for which the solution is proposed, since this paper is dealing with the application RFID for GIS technology improvements it must state the aspects of improvements
3. Introductory sentences are irrelevant, with many technical terms used without proper understanding and relevance with the subject matter ~lines 38-49.
4. Lines 50-51 what is meant by further enhancements, the research objective needs to be concrete and legible enough for the reader to study the manuscript further
5. Lines 52-58 research problem definition is not well written, please describe what type of RFID and particular GIS protocol is under consideration
6. For technical papers we do not use the term commendable, please indicate numbers ~Line 70
7. The entire related work section is irrelevant, please consider your research problem and then write the state-of-the-art. Terms i.e., J2EE application, ArcGIS Server servers, Struts2, etc. are neither defined nor cited

Experimental design

1.The proposed system is a three-layered system, which layer utilizes LANDMARK
2. Equations are not cited
3. The UWB Positioning and Tracking dataset needs a formal introduction and reference to the source
4. Section 4.3 is inconsistent with the introductory section, especially the research objectives
5. Even the conclusion is not consistent with section 4.3.

Validity of the findings

Yes valid.

Additional comments

After revisions, the paper will be accepted.

Reviewer 2 ·

Basic reporting

See below

Experimental design

See below

Validity of the findings

See below

Additional comments

1. Add a paragraph about the research objectives, research questions and followed by the proposed method eventually organization of contents in the manuscript
2. Equations needs to be cited, moreover these equations needs a little more explanations
3. State of the art is not enough adding more references will be beneficial
4. The related work section starting paragraph is based on irrelevant claim ~line 85-88
5. The ending paragraph of related work is not fulfilling the purpose, it should be rephrased or another additional paragraph may be added ~line 122-135
6. The Line 334 needs reference to the dataset i.e., UWB dataset
7. How specifically you can use data for indoor environment testing while most of the UWB dataset is for outdoor environment
8. Please indicate empirically the improvements ~line 352-357
9. A good suggestion for the work is to identify the system capabilities in terms of number of users, as number of users increases the latency increase (Insignificantly) therefore authors may identify the upper limit of the users ~line 413.
10. The concluding sentences are describing well about the system capabilities and it is recommended to indicates the limitations of the system for particular use cases.
11. Discussion section is well written but it is advised to add a few details about the specification of the system and its potential scalability areas.
12. Some language and grammar issues may be rectified
13. Authors may add section or subsection for formal introduction of LANDMARC and other similar algorithms for better understandings of the readers.

---

## Round 0.2 · accepted · Accept

Dear authors

We have now received feedback from the experts on your revised manuscript. Bases on their input, I'm pleased to inform you that your manuscript now qualifies for the acceptance criteria.

Congratulations and thank you for your valuable contribution

Reviewer 1 ·

Basic reporting

Yes, the basic report is good for convincing the idea.

Experimental design

The experimental design is good for new ideas.

Validity of the findings

Yes, Valid.

Additional comments

Grammatical mistakes must be checked before publication.

Reviewer 2 ·

Basic reporting

This paper has been well revised.

Experimental design

This paper has been well revised.

Validity of the findings

This paper has been well revised.

Additional comments

This paper has been well revised.